# OUT OF THE SHADOWS: EXPLORING A LATENT SPACE FOR NEURAL NETWORK VERIFICATION

**Lukas Koller**[1]   **Tobias Ladner**[1]   **Matthias Althoff**[1]
[1]Technical University of Munich
`{lukas.koller, tobias.ladner, althoff}@tum.de`

## ABSTRACT

Neural networks are ubiquitous. However, they are often sensitive to small input changes. Hence, to prevent unexpected behavior in safety-critical applications, their formal verification – a notoriously hard problem – is necessary. Many state-of-the-art verification algorithms use reachability analysis or abstract interpretation to enclose the set of possible outputs of a neural network. Often, the verification is inconclusive due to the conservatism of the enclosure. To address this problem, we propose a novel specification-driven input refinement procedure, i.e., we iteratively enclose the preimage of a neural network for all unsafe outputs to reduce the set of possible inputs to only enclose the unsafe ones. For that, we transfer output specifications to the input space by exploiting a latent space, which is an artifact of the propagation of a projection-based set representation through a neural network. A projection-based set representation, e.g., a zonotope, is a "shadow" of a higher-dimensional set – a latent space – that does not change during a set propagation through a neural network. Hence, the input set and the output enclosure are "shadows" of the same latent space that we can use to transfer constraints. We present an efficient verification tool for neural networks that uses our iterative refinement to significantly reduce the number of subproblems in a branch-and-bound procedure. Using zonotopes as a set representation, unlike many other state-of-the-art approaches, our approach can be realized by only using matrix operations, which enables a significant speed-up through efficient GPU acceleration. We demonstrate that our tool achieves competitive performance compared to the top-ranking tools of the international neural network verification competition.

## 1 INTRODUCTION

Neural networks perform exceptionally well across many complex tasks, e.g., object detection (Zou et al., 2023) or protein structure prediction (Jumper et al., 2021). However, neural networks can be sensitive towards small input changes, e.g., often adversarial attacks can provoke misclassifications (Goodfellow et al., 2015). Thus, neural networks must be formally verified to avoid unexpected behavior in safety-critical applications, e.g., autonomous driving (Chib & Singh, 2023) or airborne collision avoidance (Irfan et al., 2020), where the inputs can be influenced by sensor noise or external disturbances.

The goal of formal verification of neural networks is to find a mathematical proof that every possible output for a given input set is safe with respect to a given specification; in this work, we demand that the output avoids an unsafe set. The verification problem is undecidable in the general case (Ivanov et al., 2019) and NP-complete for neural networks with ReLU-activation functions (Katz et al., 2017, Appendix I). Many prominent verification algorithms use reachability analysis or abstract interpretation to enclose the intersection of an unsafe set with the output of the neural network (Gehr et al., 2018; Singh et al., 2019a; Xu et al., 2021; Bak, 2021; Kochdumper et al., 2023): The input set is represented using a continuous set representation (such as intervals or zonotopes), which is conservatively propagated through the layers of a neural network to enclose all possible outputs. If the intersection of the output enclosure with an unsafe set is empty, the safety of the given input set is formally verified. Out of the box, most reachability-based algorithms do not scale well to large neural networks with high-dimensional input spaces because the conservatism of the set propagation increases due to over-approximating nonlinearities in the neural network. Most verification

algorithms reduce the conservatism by integrating a branch-and-bound procedure to recursively split the verification problem into smaller and simpler subproblems, e.g., by exploiting the piecewise linearity of a $\mathrm{ReLU}$-activation function to reduce approximation errors (Bunel et al., 2018) or splitting the input set to reduce its size. However, in the worst case, the verification problem is split into exponentially many subproblems, all of which must be verified.

In this paper, we speed up the formal verification of neural networks by iteratively refining the input set to only enclose the unsafe inputs. Thereby, we reduce the size of the input set to reduce the number of splits and, ultimately, the number of subproblems to be verified. For our iterative input refinement, we exploit a latent space to transfer the unsafe set backwards through the network from the output space to the input space for the enclosure of all unsafe inputs, i.e., we can discard all inputs that are already proven to be safe. The latent space is an artifact of using projection-based set representations, which represent the projection ("shadow") of a higher-dimensional set, e.g., a zonotope is the "shadow" of a higher-dimensional hypercube. The set propagation through a neural network only changes the "shadow", while the higher-dimensional set remains unchanged. Hence, all considered sets are different "shadows" of the same higher-dimensional set, representing a latent space. Thus, we can exploit the dependencies between the considered sets through the latent space to transfer the unsafe set from the output space to the input space, thereby refining the input set. Moreover, if we cannot verify the safety of the neural network, we can utilize the latent space to generate candidates for counterexamples. Ultimately, we propose a verification algorithm that integrates our novel iterative input refinement into a branch-and-bound procedure for verifying and falsifying neural networks. Further, unlike many other state-of-the-art verification algorithms, we implement our verification algorithm using only matrix operations to take full advantage of batch-wise computations and GPU acceleration to simultaneously verify entire batches of subproblems.

To summarize, our main contributions are:

- A novel iterative specification-driven input refinement procedure to speed up branch-and-bound neural network verification and falsification that exploits dependencies between the input and output space through a latent space.

- An efficient algorithm for neural network verification that takes full advantage of GPU acceleration by only using matrix operations and batch-wise computations.

- An extensive evaluation and comparison with state-of-the-art neural network verification tools on benchmarks from the neural network verification competition 2024 (VNN-COMP'24) (Brix et al., 2024). Additionally, we conduct extensive ablation studies to justify the design choices of our verification algorithm.

## 2 PRELIMINARIES

### 2.1 NOTATION

Lowercase letters denote vectors and uppercase letters denote matrices. We denote the $i$-th entry of a vector $x$ by $x_{(i)}$ and the entry in the $i$-th row and the $j$-th column of a matrix $A \in \mathbb{R}^{n \times m}$ by $A_{(i,j)}$. The $i$-th row is written as $A_{(i,\cdot)}$ and the $j$-th column as $A_{(\cdot,j)}$. The identity matrix is denoted by $I_n \in \mathbb{R}^{n \times n}$. We write the matrix (with appropriate size) that contains only zeros or ones as $\mathbf{0}$ and $\mathbf{1}$. Given two matrices $A \in \mathbb{R}^{m \times n_1}$ and $B \in \mathbb{R}^{m \times n_2}$, their (horizontal) concatenation is denoted by $[A\ B] \in \mathbb{R}^{m \times (n_1 + n_2)}$. The operation $\mathrm{Diag}(v)$ returns a diagonal matrix with the entries of vector $v$ on its diagonal, and the operation $|A|$ takes the elementwise absolute value of a matrix $A \in \mathbb{R}^{n \times m}$. We denote sets by uppercase calligraphic letters. Given two sets $\mathcal{S}_1, \mathcal{S}_2 \subset \mathbb{R}^n$, we write their Minkowski sum as $\mathcal{S}_1 \oplus \mathcal{S}_2 = \{s_1 + s_2 \mid s_1 \in \mathcal{S}_1, s_2 \in \mathcal{S}_2\}$. For $n \in \mathbb{N}$, $[n] = \{1, 2, \ldots, n\}$ is the set of all natural numbers up to $n$. The $n$-dimensional interval $\mathcal{I} \subset \mathbb{R}^n$ with bounds $l, u \in \mathbb{R}^n$ is written as $\mathcal{I} = [\underline{x}, \overline{x}]$, where $\forall i \in [n]\colon \underline{x}_{(i)} \leq \overline{x}_{(i)}$. For a function $f\colon \mathbb{R}^n \to \mathbb{R}^m$, we abbreviate the image of $\mathcal{S} \subset \mathbb{R}^n$ with $f(\mathcal{S}) = \{f(s) \mid s \in \mathcal{S}\}$.

### 2.2 FEED-FORWARD NEURAL NETWORKS

A (feed-forward) neural network $\Phi\colon \mathbb{R}^{n_0} \to \mathbb{R}^{n_\kappa}$ is a sequence of $\kappa \in \mathbb{N}$ layers. Each layer applies an affine map (linear layer) or an element-wise nonlinear activation function (nonlinear layer).

(a) (Constrained) Zonotope          (b) (Constrained) Zonotope as "Shadow"

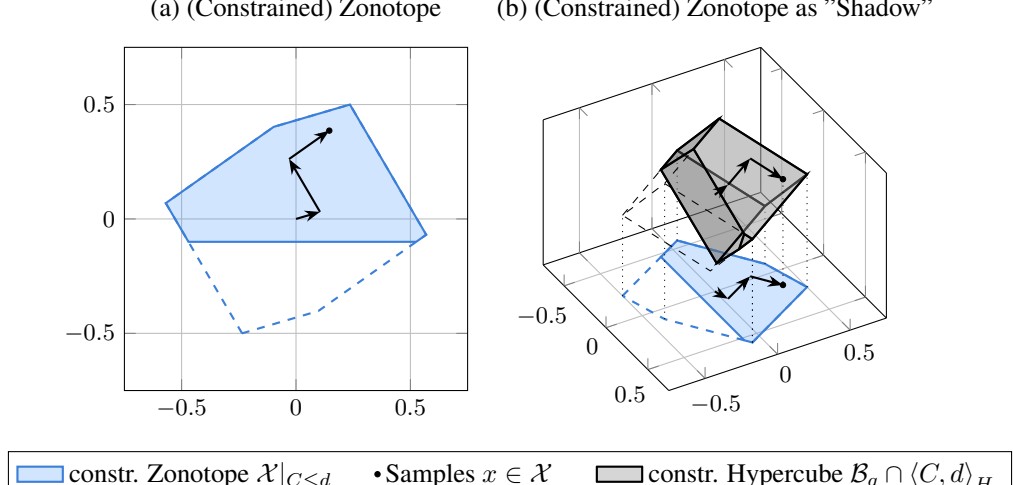

constr. Zonotope $\mathcal{X}|_{C \le d}$     • Samples $x \in \mathcal{X}$     constr. Hypercube $\mathcal{B}_q \cap \langle C, d \rangle_H$

Figure 1: (a) An illustration of a constrained zonotope and a sample with its generators; (b) the same constrained zonotope as the "shadow" of a constrained hypercube.

**Definition 1** (Neural Network, (Bishop, 2006, Sec. 5.1)). *For an input $x \in \mathbb{R}^{n_0}$, the output of a neural network $y = \Phi(x) \in \mathbb{R}^{n_\kappa}$ is*

$$h_0 = x, \qquad h_k = L_k(h_{k-1}) \quad \text{for } k \in [\kappa], \qquad y = h_\kappa,$$

*where*

$$L_k(h_{k-1}) = \begin{cases} W_k\, h_{k-1} + b_k & \text{if $k$-th layer is linear,} \\ \phi_k(h_{k-1}) & \text{otherwise,} \end{cases}$$

*with weights $W_k \in \mathbb{R}^{n_k \times n_{k-1}}$, bias $b_k \in \mathbb{R}^{n_k}$, and (element-wise) nonlinear function $\phi_k$.*

### 2.3 SET-BASED COMPUTING

A zonotope is a convex set representation popular in reachability analysis due to its favorable computational complexity (Singh et al., 2019b; Kochdumper et al., 2023).

**Definition 2** (Zonotope, (Girard, 2005, Def. 1)). *Given a center $c \in \mathbb{R}^n$ and a generator matrix $G \in \mathbb{R}^{n \times q}$, a zonotope is defined as*

$$\mathcal{Z} = \langle c, G \rangle_Z := \left\{ c + \sum_{i=1}^{q} G_{(\cdot, i)}\, \beta_{(i)} \,\middle|\, \beta \in [-1, 1]^q \right\}.$$

Subsequently, we define the set-based operations for zonotopes required for our verification algorithm. The Minkowski sum of a zonotope $\mathcal{Z} = \langle c, G \rangle_Z \subset \mathbb{R}^n$ and an interval $[\underline{x}, \overline{x}] \subset \mathbb{R}^n$ is computed by (Althoff, 2010, Prop. 2.1 & Sec. 2.4)

$$\mathcal{Z} \oplus [\underline{x}, \overline{x}] = \langle c + 1/2\,(\overline{x} + \underline{x}), [G\, \mathrm{diag}(1/2\,(\overline{x} - \underline{x}))] \rangle_Z. \tag{1}$$

The image of a zonotope $\mathcal{Z} = \langle c, G \rangle_Z \subset \mathbb{R}^n$ under an affine map $f \colon \mathbb{R}^n \to \mathbb{R}^m$, $x \mapsto W\, x + b$ with $W \in \mathbb{R}^{m \times n}$ and $b \in \mathbb{R}^m$ is computed by (Althoff, 2010, Sec. 2.4)

$$f(\mathcal{Z}) = W\, \mathcal{Z} \oplus b = \langle W\, c + b, W\, G \rangle_Z. \tag{2}$$

Using an affine map, we can write a zonotope $\mathcal{Z} = \langle c, G \rangle_Z \subset \mathbb{R}^n$ with $q$ generators, i.e., $G \in \mathbb{R}^{n \times q}$, as the projection of a $q$-dimensional (unit)-hypercube $\mathcal{B}_q = [-1, 1]^q$: $\mathcal{Z} = c \oplus G\, \mathcal{B}_q$. Thus, intuitively, a zonotope is the "shadow" of a higher-dimensional hypercube.

A convex polytope is the intersection of a finite number of halfspaces (Althoff, 2010, Def. 2.1); we denote a polytope by $\langle A, b \rangle_H = \{x \in \mathbb{R}^n \mid A\, x \le b\} \subset \mathbb{R}^n$, where $A \in \mathbb{R}^{n \times p}$ and $b \in \mathbb{R}^p$.

All zonotopes are convex and point-symmetric. However, by constraining the hypercube, we can represent arbitrary convex polytopes (Scott et al., 2016, Thm. 1).

**Definition 3** (Constrained Zonotope, (Scott et al., 2016, Def. 3)). *Given a zonotope $\mathcal{Z} = \langle c, G \rangle_Z \subset \mathbb{R}^n$ with $c \in \mathbb{R}^n$ and $G \in \mathbb{R}^{n \times q}$, the zonotope with constraints $C \beta \leq d$, for $\beta \in \mathcal{B}_q$ with $C \in \mathbb{R}^{p \times q}$ and $d \in \mathbb{R}^p$, is defined as*

$$\mathcal{Z}|_{C \leq d} := \{c + G \beta \mid \beta \in [-1, 1]^q, C \beta \leq d\} = c \oplus G \left( \mathcal{B}_q \cap \langle C, d \rangle_H \right).$$

Compared to (Scott et al., 2016, Def. 3), we use inequality constraints instead of equality constraints for convenience. Both types of constraints are equivalent and can be translated by introducing slack variables. Moreover, the considered set-based operations, i.e., Minkowski sum (1) and affine map (2), are identical for zonotopes and constrained zonotope (Scott et al., 2016, Prop. 1). Fig. 1 illustrates a constrained zonotope as a "shadow" of a constrained hypercube.

## 2.4 FORMAL VERIFICATION OF NEURAL NETWORKS

The output set of a neural network can be enclosed by conservatively propagating a zonotope through the layers of the neural network.

**Proposition 1** (Set Propagation, (Ladner & Althoff, 2023, Sec. 2.4)). *Given a neural network $\Phi \colon \mathbb{R}^{n_0} \to \mathbb{R}^{n_\kappa}$ and an input set $\mathcal{X} \subset \mathbb{R}^{n_0}$, an enclosure $\mathcal{Y} = \mathtt{enclose}(\Phi, \mathcal{X}) \subset \mathbb{R}^{n_\kappa}$ of the image $\mathcal{Y}^* := \Phi(\mathcal{X}) \subseteq \mathcal{Y}$ can be computed as*

$$\mathcal{H}_0 = \mathcal{X}, \qquad \mathcal{H}_k = \mathtt{enclose}(L_k, \mathcal{H}_{k-1}) \quad \text{for } k \in [\kappa], \qquad \mathcal{Y} = \mathcal{H}_\kappa.$$

The operation $\mathtt{enclose}(L_k, \mathcal{H}_{k-1})$ encloses the image of the $k$-th layer for the input set $\mathcal{H}_{k-1}$, i.e., $L_k(\mathcal{H}_{k-1}) \subseteq \mathtt{enclose}(L_k, \mathcal{H}_{k-1})$ (Ladner & Althoff, 2023, Prop. 2.14). If the $k$-th layer is linear, an affine map is applied (2): $\mathtt{enclose}(L_k, \mathcal{H}_{k-1}) = W_k \mathcal{H}_{k-1} \oplus b_k$; otherwise, the activation function $\phi_k$ is enclosed with a linear function and corresponding approximation errors: $\mathtt{enclose}(L_k, \mathcal{H}_{k-1}) = \mathrm{diag}(m_k) \mathcal{H}_{k-1} \oplus [\underline{e}_k, \overline{e}_k]$, where $m_{k(i)}$ is the approximation slope and $[\underline{e}_{k(i)}, \overline{e}_{k(i)}]$ the approximation error of the $i$-th neuron, with $i \in [n_k]$ (Koller et al., 2025, Sec. IV). Please note that Prop. 1 can handle arbitrary elementwise activation functions (e.g., sigmoid or hyberbolic tangent) and is not limited ReLU-activations.

## 2.5 PROBLEM STATEMENT

Given a neural network $\Phi \colon \mathbb{R}^{n_0} \to \mathbb{R}^{n_\kappa}$, an input set $\mathcal{X} \subset \mathbb{R}^{n_0}$, and an unsafe set $\mathcal{U} \subset \mathbb{R}^{n_\kappa}$, our goal is to derive an efficient and practical algorithm that can either formally verify the safety of the neural network, i.e.,

$$\Phi(\mathcal{X}) \cap \mathcal{U} = \emptyset, \tag{3}$$

or find a counterexample, i.e.,

$$x \in \mathcal{X} \text{ such that } \Phi(x) \in \mathcal{U}. \tag{4}$$

# 3 SPECIFICATION-DRIVEN INPUT REFINEMENT

A zonotope is the "shadow" of a higher-dimensional hypercube (Fig. 1b). The propagation of a zonotope through the layers of a neural network transforms the projection of the hypercube, but the hypercube itself remains unchanged. Thus, all enclosed sets, i.e., the input, hidden, and output sets, are different "shadows" of the same hypercube. Therefore, the hypercube represents a latent space. Let us demonstrate the "shadow" view by an example:

**Example 1.** *Fig. 2 illustrates the propagation of a two-dimensional input set $\mathcal{X}$ through a linear layer and a $\mathrm{ReLU}$-activation function, i.e., $x \mapsto \mathrm{ReLU}(2^{-1/2} \left[\begin{smallmatrix} 1 & -1 \\ 1 & 1 \end{smallmatrix}\right] x + \left[\begin{smallmatrix} 1 \\ 0 \end{smallmatrix}\right])$. Intuitively, the linear layer rotates the hypercube, and the nonlinear layer tilts it to compute the output set as its "shadow."*

*For input set $\mathcal{X} = \left\langle \mathbf{0}, 2^{-1/2} I_2 \right\rangle_Z$, the hidden set $\mathcal{H}_1$ and the $\mathcal{Y}$ are computed using Prop. 1:*

$$\mathcal{H}_1 = 2^{-1/2} \begin{bmatrix} 1 & -1 \\ 1 & 1 \end{bmatrix} \mathcal{X} + \begin{bmatrix} 1 \\ 0 \end{bmatrix} = \left\langle \begin{bmatrix} 1 \\ 0 \end{bmatrix}, \,^{1}\!/_2 \begin{bmatrix} 1 & -1 \\ 1 & 1 \end{bmatrix} \right\rangle_Z,$$

$$\mathcal{Y} = \mathrm{diag}\left( \begin{bmatrix} 1 \\ ^{1}\!/_2 \end{bmatrix} \right) \mathcal{H}_1 + \left[ \begin{bmatrix} 0 \\ 0 \end{bmatrix}, \begin{bmatrix} 0 \\ ^{1}\!/_2 \end{bmatrix} \right] = \left\langle \begin{bmatrix} 1 \\ ^{1}\!/_4 \end{bmatrix}, \,^{1}\!/_4 \begin{bmatrix} 2 & -2 & 0 \\ 1 & 1 & 1 \end{bmatrix} \right\rangle_Z. \tag{5}$$

(a) Input Space (b) Hidden Space (c) Output Space

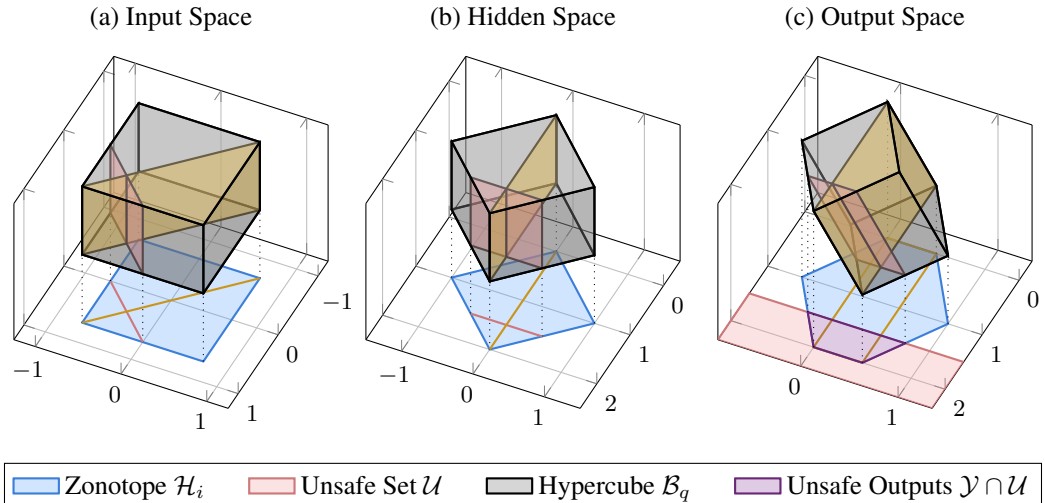

| Zonotope $\mathcal{H}_i$ | Unsafe Set $\mathcal{U}$ | Hypercube $\mathcal{B}_q$ | Unsafe Outputs $\mathcal{Y} \cap \mathcal{U}$ |

Figure 2: Illustrating the zonotope propagation through a linear layer and a nonlinear layer as the "shadows" of the same hypercube: (a) The input set in the input space; (b) The output of the linear layer, which rotates and offsets the input set; (c) The output of the nonlinear layer tilts the hypercube to add the approximation errors. Using the hypercube, the unsafe set (□) is transferred from the output space to the input space. Additionally, the hypercube is split along a hyperplane (□) to exploit the piecewise linearity of the ReLU-activation function.

*The approximation slope and errors for enclosing the output of the* ReLU*-activation function are computed using (Koller et al., 2025, Prop. 10). For each input $x \in \mathcal{X}$ and its corresponding output $y = \Phi(x) \in \mathcal{Y}$, we use the definition of a zonotope (Def. 2) to obtain $\beta \in [-1, 1]^3$ such that*

$$x = \begin{bmatrix} 1/\sqrt{2} & 0 \\ 0 & 1/\sqrt{2} \end{bmatrix} \beta_{([2])}, \qquad y = \begin{bmatrix} 1 \\ 1/4 \end{bmatrix} + \begin{bmatrix} 1/2 & -1/2 \\ 1/4 & 1/4 \end{bmatrix} \beta_{([2])} + \begin{bmatrix} 0 \\ 1/4 \end{bmatrix} \beta_{(3)}. \tag{6}$$

*The input $x$ and the output $y$ are represented using the same factors $\beta_{([2])}$ with an additional factor $\beta_{(3)}$ for the approximation error. We can use an unsafe set $y_{(1)} \geq 3/2$ to formluate constraints (□ in Fig. 2) on the factors $\beta_{([2])}$ of the input:*

$$y_{(1)} \geq 3/2 \overset{(6)}{\iff} 1 + [1/2 \quad -1/2] \beta_{([2])} + 0 \, \beta_{(3)} \geq 3/2 \iff \underbrace{[-1/2 \quad 1/2]}_{= C} \beta_{([2])} \leq \underbrace{-1/2}_{= d}. \tag{7}$$

*Therefore, we can use the constraints on $\beta$, i.e., $C\beta \leq d$, to refine the input set to $\mathcal{X}|_{C \leq d} \subseteq \mathcal{X}$, which encloses all unsafe inputs. Analogously, we can split the set along $h_1 \leq \bar{0} \iff [1/2 \quad 1/2] \beta_{([2])} \leq 0$, for $h_1 \in \mathcal{H}_1$, to exploit the piecewise linearity of the ReLU-activation function (□ in Fig. 2).*

As example 1 demonstrates, we can constrain the input set with an unsafe set in the output space by exploiting the dependencies between the considered sets through the latent space. Prop. 2 formalizes an input refinement based on our observations.

**Proposition 2** (Enclosing Unsafe Inputs). *Given are a neural network $\Phi \colon \mathbb{R}^{n_0} \to \mathbb{R}^{n_\kappa}$, an input set $\mathcal{X} = \langle c_x, G_x \rangle_Z \subset \mathbb{R}^{n_0}$ with $G_x \in \mathbb{R}^{n_0 \times q_0}$, and an unsafe set $\mathcal{U} = \langle A, b \rangle_H \subset \mathbb{R}^{n_\kappa}$. Let $\mathcal{Y} = \langle c_y, G_y \rangle_Z = \texttt{enclose}(\Phi, \mathcal{X})$ be an enclosure of the output set with $G_y \in \mathbb{R}^{n_\kappa \times q_\kappa}$. We enclose all unsafe inputs by constraining $\mathcal{X}$ to $\mathcal{X}|_{C \leq d}$, i.e.,*

$$\{x \in \mathcal{X} \mid \Phi(x) \in \mathcal{U}\} \subseteq \mathcal{X}|_{C \leq d} \subseteq \mathcal{X},$$

*where $C := A \, G_{y(\cdot, [q_0])}$ and $d := b - A \, c_y + \left| A \, G_{y(\cdot, [q_\kappa] \setminus [q_0])} \right| \mathbf{1}$.*

*Proof.* See Appendix F.

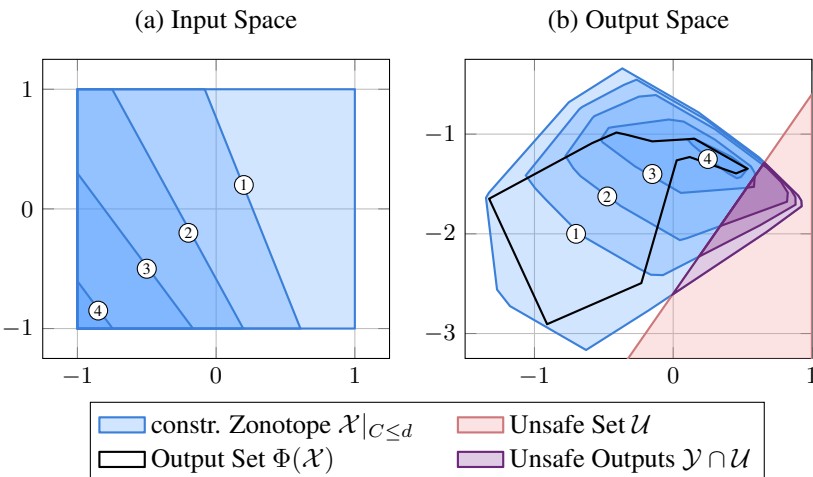

Figure 3: Illustration of an iterative refinement of the input set (Prop. 2): After the fourth iteration, the intersection with the unsafe set is empty, and the neural network is verified.

We can iteratively apply Prop. 2 to refine the input set to minimize the intersection of the computed output set and the unsafe set. During each iteration, we transfer the unsafe set from the output space to the input space to remove the parts of the input set that are provably safe. Please note that our input refinement only encloses all unsafe inputs and does not, in general, preserve an enclosure of the full output set. In Fig. 3, we iterate Prop. 2 until the intersection of the output set and the unsafe set is empty, and thus the problem is verified.

## 4 SET-BASED FALSIFICATION, VERIFICATION, AND INPUT REFINEMENT

For the fast and practical verification of neural networks, we integrate our novel input refinement (Prop. 2) into a branch-and-bound procedure (Alg. 1), where we utilize the enclosed output set for verification, falsification, and input refinement. In each iteration, we perform the following computations: (i) The output set of the current input set is enclosed (Prop. 1). (ii) The intersection with the unsafe set is checked. (iii) If the intersection is non-empty, the verification is inconclusive, and falsification is attempted. (iv) Finally, if we cannot verify the input set nor find a counterexample, we split and refine the input set (Prop. 2). The subsequent subsections describe the steps of Alg. 1 in detail.

**Set-Based Verification** We compute the output set $\mathcal{Y} = \langle c_y, G_y \rangle_Z \subset \mathbb{R}^{n_\kappa}$ of the current input set using Prop. 1 and check if the intersection with the unsafe set $\mathcal{U} = \langle A, b \rangle_H \subset \mathbb{R}^{n_\kappa}$ with $A \in \mathbb{R}^{p \times n_\kappa}$ is empty (Scott et al., 2016, Prop. 2):

$$\exists i \in [p]\colon A_{(i,\cdot)}\, c_y - \left| A_{(i,\cdot)}\, G_y \right| \mathbf{1} > b_{(i)} \implies \mathcal{Y} \cap \mathcal{U} = \emptyset. \tag{8}$$

**Set-Based Falsification** If we cannot verify the current input set, we try to find a counterexample within it. For that, we utilize the latent space (Sec. 3) to identify the input for a boundary point of the intersection of the enclosed output set and the unsafe set. For each normal vector of the unsafe set specification $A_{(i,\cdot)} \in \mathbb{R}^{n_\kappa}$, for $i \in [p]$, we compute a boundary point (Althoff & Frehse, 2016, Lemma 1):

$$\tilde{\beta}_i = \text{sign}\!\left(A_{(i,\cdot)}\, G_y\right), \qquad\qquad \tilde{y} = c_y + G_y\, \tilde{\beta}_i. \tag{9}$$

Finally, we check if the corresponding input $\tilde{x} = c_x + G_x\, \tilde{\beta}_{i([q_0])} \in \mathcal{X}$ in the input set $\mathcal{X} = \langle c_x, G_x \rangle_Z$ with $G_x \in \mathbb{R}^{n_0 \times q_0}$ is a counterexample, i.e., whether $\Phi(\tilde{x}) \in \mathcal{U}$.

**Input Refinement and Splitting** We reduce the conservatism of the zonotope propagation by splitting the input set before applying our input refinement (Prop. 2). The splitting is realized by

---

**Algorithm 1:** Set-based verification algorithm. We store input sets in a queue and use the operations `initQueue`, `isNonEmpty`, `enqueue`, and `dequeue` from (Knuth, 1997, Sec. 2.2.1)

---

**Input:** Neural network $\Phi$, input set $\mathcal{X} = \langle c_x, G_x \rangle_Z$, and unsafe set $\mathcal{U} = \langle A, b \rangle_H$
**Output:** Verification result, i.e., VERIFIED or FALSIFIED($\tilde{x}$) with counterexample $\tilde{x} \in \mathcal{X}$

1   **function** `verify`$(\Phi, \mathcal{X}, \mathcal{U})$
2     $Q \leftarrow$ `initQueue()`, $i \leftarrow 1$             // Initialize queue & counter.
3     $Q$.`enqueue`$(\mathcal{X})$                 // Add the initial input set.
4     **while** $Q$.`isNonEmpty()` **do**
5        $\mathcal{X}_i \leftarrow Q$.`dequeue()`               // Get the next input set.
6        $\mathcal{Y}_i \leftarrow$ `enclose`$(\Phi, \mathcal{X}_i)$        // Enclose the output set (Prop. 1).
7        **if** $\mathcal{Y}_i \cap \mathcal{U} = \emptyset$ **then**             // 1. Verification
8           **continue**                 // Verified input set.
9        **else**                         // 2. Falsification
10           $\tilde{x}_i \leftarrow c_x + G_x \tilde{\beta}_{([q_0])}$ for $\tilde{\beta}_i = \text{sign}\big(A_{(i,\cdot)} G_y\big)$    // Generate an adversarial input (9)
11           $\tilde{y}_i \leftarrow \Phi(x)$            // Compute the adversarial output (Def. 1).
12           **if** $\tilde{y}_i \in \mathcal{U}$ **then**
13             **return** FALSIFIED($\tilde{x}_i$)          // Found an unsafe input.
14           **else**                // 3. Refinement & Splitting
15             $C, d \leftarrow$ Compute the constraints for the input set.      // Prop. 2
16             $\mathcal{X}_i|_{C_1 \leq d_1}, \ldots, \mathcal{X}_i|_{C_\xi \leq d_\xi} \leftarrow$ Split the input set.    // (Scott et al., 2016, Prop. 3)
17             $Q$.`enqueue`$\big(\mathcal{X}_i|_{C_1 \leq d_1}, \ldots, \mathcal{X}_i|_{C_\xi \leq d_\xi}\big)$    // Add new sets to the queue.
18             $i \leftarrow i + 1$                 // Increment the counter.
19           **end**
20        **end**
21     **end**
22     **return** VERIFIED            // Queue is empty; verified all input sets.

---

adding constraints to the hypercube (Scott et al., 2016, Prop. 3); e.g., we can split the input set or we can formulate constraints that exploit the piecewise linearity of the ReLU-activation function (■ in Fig. 2). For each split, we heuristically select a dimension or a neuron by comparing gradients w.r.t. a zonotope norm of the output set (Appendix B).

## 5   EVALUATION

We have implemented our verification algorithm in MATLAB as part of the CORA toolbox (Althoff, 2015). To make our evaluation as transparent and reproducible as possible, we compare our tool on benchmarks from the neural network verification competition 2024 (VNN-COMP'24) (Brix et al., 2024) with the top-5 tools of the competition: $\alpha$-$\beta$-CROWN (Wang et al., 2021), NeuralSAT (Duong et al., 2023), PyRat (Lemesle et al., 2024), Marabou (Wu et al., 2024), and nnenum (Bak, 2021).

Tab. 1 presents the results for 8 competitive and standardized benchmarks of the VNN-COMP'24 (Brix et al., 2024). Please see Appendix A for details. Across all benchmarks, our verification algorithm achieves competitive performance, even matching the top performance in four benchmarks.

### 5.1   ABLATION STUDIES

We run extensive ablation studies to justify the different design choices of our algorithm.

**Input Refinement**    We evaluate the verification improvement and per-iteration overhead of our input refinement (Tab. 2). Therefore, (i) we compare the number of subproblems required for verifying or falsifying instances with (✓) and without (✗) our input refinement enabled: We observe that our refinement reduces the number of subproblems by 59.6% for `acasxu` and 65.3% for `safenlp`. (ii) Further, while the per-iteration time is larger with our input refinement, the maximum verification time is reduced by 53.5% for `acasxu` and 37.7% for `safenlp`.

Table 1: Main Results [*%Solved* ↑ *(#Verified / #Falsified)*], where *%Solved* is the percentage of solved instances (verified or falsified), with counts shown in parentheses.

| Tool | acasxu | collins -rul-cnn | cora | dist -shift | linear -izenn | meta -room | safenlp | tllverify -bench |
|---|---|---|---|---|---|---|---|---|
| CORA (Ours) | 99.5% (138 / 47) | **100.0%** (30 / 32) | 81.1% (18 / 128) | **98.6%** (63 / 8) | **100.0%** (59 / 1) | 97.0% (90 / 7) | 89.2% (311 / 652) | **100.0%** (15 / 17) |
| *Results from VNN-COMP'24 (Brix et al., 2024)* | | | | | | | | |
| $\alpha$-$\beta$-CROWN | **100.0%** (139 / 47) | **100.0%** (30 / 32) | **87.8%** (24 / 134) | **98.6%** (63 / 8) | **100.0%** (59 / 1) | **98.0%** (91 / 7) | **100.0%** (421 / 659) | **100.0%** (15 / 17) |
| NeuralSAT | 98.9% (138 / 46) | **100.0%** (30 / 32) | 87.2% (23 / 134) | **98.6%** (63 / 8) | **100.0%** (59 / 1) | **98.0%** (91 / 7) | 90.5% (327 / 650) | **100.0%** (15 / 17) |
| PyRAT | 98.9% (137 / 47) | 93.5% (30 / 28) | 83.3% (22 / 128) | **98.6%** (63 / 8) | **100.0%** (59 / 1) | 97.0% (91 / 6) | 79.9% (277 / 586) | **100.0%** (15 / 17) |
| Marabou | 96.2% (134 / 45) | **100.0%** (30 / 32) | 86.7% (22 / 134) | 95.8% (62 / 7) | **100.0%** (59 / 1) | 53.0% (46 / 7) | 62.5% (300 / 375) | 93.8% (13 / 17) |
| nnenum | 99.5% (139 / 46) | **100.0%** (30 / 32) | 14.4% (20 / 6 ) | – (– / –) | 98.3% (59 / 0) | 46.0% (44 / 2) | 89.2% (321 / 642) | 56.2% (2 / 16) |

Table 2: Input Refinement: without (✗) vs. with (✔).

| Input Refinement | acasxu | | safenlp | |
|---|---|---|---|---|
| | ✗ | ✔ (Ours) | ✗ | ✔ (Ours) |
| *avg. #Sub- problems* ↓ *(max)* | 1 611.2 (133 438) | **651.5** (36 134) | 4 401.8 (293 304) | **1 529.2** (114 065) |
| *%Solved* ↑ *(#Verified / #Falsified)* | **99.5%** (138 / 47) | **99.5%** (138 / 47) | 80.8 (298 / 575) | **89.2%** (311 / 652) |
| *max Time* ↓ *max Time per Iter.* ↓ | 32.5s **118.6**ms | **15.1**s 302.8ms | 19.9s **54.3**ms | **12.4**s 89.4ms |

**GPU Acceleration and Batch Size**  We demonstrate the efficacy and speed up of the GPU acceleration and batch size by comparing the results of the `acasxu` and `safenlp` benchmark computed on a CPU (Tab. 3a): The GPU acceleration enables a significant speed up, which allows the verification of more instances within the allotted timeout, i.e., the GPU speeds up the verification by 82.8% on `acasxu` and 89.8% on `safenlp`.

**Falsification Method**  We compare our set-based adversarial attack (Sec. 4) against the fast-gradient-sign method (FGSM) (Goodfellow et al., 2015) on the `safenlp` benchmark. Tab. 3b shows the number of instances falsified within the first 50 iterations to compare the falsification strength without influence from the runtime. Our set-based adversarial attack outperforms FGSM by falsifying 60.3% more instances. Further, the main results (Tab. 1) show that in most benchmarks we match the top falsification performance.

# 6 RELATED WORK

**Neural Network Verification and Adversarial Attacks**  Most algorithms for neural network verification either (i) formulate an optimization or constraint satisfaction problem and apply an off-the-shelf solver, e.g., satisfiability modulo theories (Katz et al., 2017; Duong et al., 2023; Wu et al., 2024) or (mixed-integer) linear programming (Singh et al., 2019b; Müller et al., 2022), or (ii) use abstract interpretation or reachability analysis to enclose the intersection of the specification with the output of the neural network (Gehr et al., 2018; Wang et al., 2018; Singh et al., 2019a; Wang et al., 2021; Kochdumper et al., 2023; Ladner & Althoff, 2023; Lemesle et al., 2024).

The most common abstract domains or set representations are intervals (Gehr et al., 2018), zonotopes (Gehr et al., 2018), or polytopes (Müller et al., 2022; Zhang et al., 2018; Singh et al., 2019a),

Table 3: Ablation studies.

(a) GPU vs. CPU [*max. Verification Time* ↓ *(%Solved)*].

| **Method** *(Batch Size)* | acasxu | safenlp |
|---|---|---|
| CPU (1) | 40.1s (95.7%) | 14.7s (83.1%) |
| GPU (128) | 7.3s (99.5%) | 2.0s (86.8%) |
| GPU (1024) | **6.9**s (99.5%) | **1.5**s (89.2%) |

(b) Falsification [*#Falsified* ↑ *(%Solved)*].

| **Falsification** | safenlp (≤ 50 *iter.*) |
|---|---|
| FGSM | 257 (23.8%) |
| Ours | **647** (59.9%) |

that use linear relaxations of activation functions for the propagation. More complex set representations can enclose the output set more tightly, e.g., polynomial zonotopes (Kochdumper et al., 2023), hybrid zonotopes (Ortiz et al., 2023), star sets (Bak, 2021); however, due to higher computational cost, these approaches do not scale to large neural networks. The results of multiple abstract domains can be combined to obtain a tighter enclosure (Lemesle et al., 2024). A popular abstract domain, DeepPoly/CROWN (Singh et al., 2019a; Zhang et al., 2018), uses bounded polytopes, which are represented as the intersection of linear bounds and intervals. Further, gradient-based optimization can be applied to bounding parameters to tighten the computed enclosure (Wang et al., 2021).

If formal verification is computationally infeasible, an alternative is to falsify neural networks using adversarial attacks. Often, gradient-based attacks are fast and effective at finding adversarial perturbations (Goodfellow et al., 2015; Kurakin et al., 2017). Further, stronger adversarial examples can be computed with optimization-based approaches (Carlini & Wagner, 2017). Our set-based falsification does not require the gradient of the neural network, while being faster to compute than an optimization problem.

**Iterative Neural Network Refinement**  There are different iterative refinement approaches, e.g., refining bounds using (mixed-integer) linear programs (Wang et al., 2018; Singh et al., 2019b; Yang et al., 2021), counterexample guided abstraction refinement (Wu et al., 2024), and refining the hyperparameters of output set enclosure (Xu et al., 2021; Ladner & Althoff, 2023). Our novel specification-driven input refinement iteratively encloses the set of inputs that cause an intersection with an unsafe set in the output space. Therefore, our refinement does not require tuning hyperparameters or solving expensive (mixed-integer) linear programs. The refinement procedure uses dependencies between propagated zonotopes, which can be used to simplify computations for reachability analysis or identify falsifying states (Kochdumper et al., 2020).

Further, our input refinement can be used to enclose the preimage of a neural network. Linear bounding techniques can be used to enclose preimages by solving multiple linear programs to compute bounds with respect to constraints or dimensions separately (Kotha et al., 2023; Zhang et al., 2024). Conversely, we compute a zonotope enclosing the preimage that we optimize with an iterative procedure with respect to all constraints and dimensions simultaneously.

**Branch-and-Bound Algorithms and Splitting Heuristics**  For large neural networks, most verification approaches are too conservative. Therefore, in practice, most verification approaches use a branch-and-bound procedure that recursively splits the verification problem into smaller subproblems that are easier to solve (Brix et al., 2023; 2024), e.g., by exploiting the piece-wise linearity of the ReLU-activation function. There are various approaches and heuristics for splitting a verification problem, e.g., largest radius, largest approximation error and its effect on the output constraints (Henriksen & Lomuscio, 2021), gradient or sensitivity-based heuristics to estimate the impact of a neuron on the output (Balunovic et al., 2019; Ladner & Althoff, 2023), magnitude of the coefficient of linear relaxation (Durand et al., 2022), largest unstable neuron in the first undecidable layer (Yin et al., 2022), multi-neuron constraints (Ferrari et al., 2022), least unstable neuron (Duong et al., 2023).

## 7 CONCLUSION

In this paper, we present a novel iterative specification-driven input refinement procedure to speed up the formal verification of neural networks. For the refinement, we exploit a latent space that is an artifact from the propagation of projection-based set representations, e.g., zonotopes, through the layers of a neural network. Our procedure iteratively refines an enclosure of the unsafe inputs by using the latent space to constrain the input set with the unsafe set from the output space. We integrate our refinement procedure into a branch-and-bound neural network verification algorithm. In an extensive evaluation, we show a significant reduction in the number of recursive splits required for verification. Moreover, we demonstrate that our algorithm achieves competitive performance compared to the top-5 tools from the most recent neural network verification competition. In summary, the exploitation of the proposed latent space presents a promising new direction for the formal verification of neural networks.

## ACKNOWLEDGEMENTS

This work was partially supported by the project SPP-2422 (No. 500936349) and the project FAI (No. 286525601), both funded by the German Research Foundation (DFG).

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

## A  APPENDIX – EVALUATION DETAILS

For the evaluation, we implement our verification algorithm in MATLAB using the CORA toolbox (Althoff, 2015).

**Hardware**  To avoid an unfair comparison with the results of the VNN-COMP'24 (Brix et al., 2024), we run all our evaluations on a laptop with inferior hardware compared to the VNN-COMP'24, i.e., Intel Core i7-13700H and a NVIDIA RTX 4070 Laptop GPU (8GB). The competition utilizes servers equipped with an NVIDIA A10G GPU, featuring significantly more memory (24GB) (Brix et al., 2024, Sec. 2).

**VNN-COMP** The selected benchmarks are taken from the VNN-COMP'24 (Brix et al., 2024)[1]: `acasxu` a standard benchmark used by many prior works containing neural networks for airborne collision detection; `collins-rul-cnn` containing neural networks for condition-based maintenance; `cora` and `metaroom` containing neural networks for high-dimensional image classification; `dist-shift` containing neural networks for distribution shift detection; `linearizenn` containing neural networks for an autonomous aircraft taxiing system; `safenlp` containing neural networks for sentence classification; `tllverifybench` containing two-level lattice neural networks.

Our selection of the top-5 tools is based on the ranking after the tool updates that resolved many penalties due to incorrectly formatted counterexamples (Brix et al., 2024, Tab. 35).

**Ablation Studies** To make a meaningful comparison in our ablation studies, we compute the statistics in Tab. 2, 3a and 5 only for instances that all considered comparands have solved, because the additional instances solved would inflate the number of verified subproblems or the number of iterations, which would skew the statistics and prevent a meaningful comparison.

**Refinement Termination** We terminate our input refinement after a fixed number of iterations. In each iteration of Alg. 1, we apply 8 refinement iterations (Prop. 2) to the split input sets. For each refinement iteration, we apply our constraint zonotope bounding procedure (Prop. 3) for at most 4 iterations.

**Numerical Tolerance** All counterexamples are checked with respect to the numerical tolerance specified in the rules of the VNN-COMP (Brix et al., 2024, Sec. 2). In our experiments, we did not observe any additional numerical instability caused by our input refinement.

## B  APPENDIX – ENCLOSURE-GRADIENT SPLITTING HEURISTIC

In each iteration of our verification algorithm (Alg. 1), we either split each input set along at center of an input dimension or split the input set to an unstable ReLU-neuron along 0 to exploit the piece-wise linearity of ReLU. Typical gradient-based splitting heuristics use the local gradient of the neural network to rank the importance of different input dimensions or neurons (Balunovic et al., 2019; Ladner & Althoff, 2023). The local gradient does not explicitly consider effects of splitting input dimensions or neurons on the conservatism of the enclosure, i.e., a split does not guarantee a tighter enclosure. Therefore, we compute the gradient of the size of the output set with respect to the enclosure. We use the F-radius to measure the size of the output enclosure $\mathcal{Y} = \langle c_y, G_y \rangle_Z = \texttt{enclose}(\Phi, \mathcal{X})$, i.e., the F-radius is defined as the Frobenius norm of the generator matrix (Combastel, 2015, Def. 3): $\|\mathcal{Y}\|_F := \sqrt{\sum_{i=1}^{n} \sum_{j=1}^{q} G_{y(i,j)}^2}$. Intuitively, we use the gradient of the F-radius to measure the contribution of an input dimension or an approximation error to the overall size of the output enclosure. For an input dimension $i \in [n_0]$, we compute the score

$$s(i) = r_{(i)} \nabla_{r_{(i)}} \|\mathcal{Y}\|_F, \tag{10}$$

where the $r = |G_x| \mathbf{1}$ is the radius of the input set $\mathcal{X} = \langle c_x, G_x \rangle_Z \subset \mathbb{R}^{n_0}$. If the $k$-th layer is a ReLU-activation layer, for $k \in [\kappa]$, we compute the score for the $i$-th neuron, for $i \in [n_k]$, by

$$s(L_k, i) = e_{k(i)} \nabla_{e_{k(i)}} \|\mathcal{Y}\|_F, \tag{11}$$

where $e_k = \frac{1}{2}(\overline{e}_k - \underline{e}_k)$ is the radius of the approximation error. In each iteration, we split the input dimension or the neuron with the largest score.

In Tab. 4, we compare our enclosure-gradient heuristic against a local-gradient heurtistic defined as (Ladner & Althoff, 2023)

$$s(i) = r_{(i)} \|\nabla_{r_{(i)}} y\|_2 \qquad\qquad s(L_k, i) = e_{k(i)} \|\nabla_{e_{k(i)}} y\|_2, \tag{12}$$

where $y \in \mathbb{R}^{n_\kappa}$ is the output of the neural network for the center of the input set, i.e., $y = \Phi(c_x)$. Our enclosure-gradient heuristic outperforms the local-gradient heuristic on the considered benchmarks (`acasxu` and `safenlp`), demonstrating its effectiveness.

---

[1] At the time of writing, VNN-COMP'24 represented the most recent benchmark suite; the finalized benchmarks and results of VNN-COMP'25 were released after the submission of this manuscript.

Table 4: Splitting Heuristics.

| Heuristic
+ Input Refinement Prop. 2 | %Solved ↑
(#Verified / #Falsified) | avg. #Sub-
problems ↓ (max) | avg. Time
↓ (max) |
|---|---|---|---|
| `acasxu` | | | |
| Local Gradient | 98.4%
(136 / 47) | 8 580.5
(430 403) | 1.8s
(85.2s) |
| Enclosure Gradient (Ours) | **99.5**%
(138 / 47) | **620.3**
(36 134) | **0.8**s
(15.1s) |
| `safenlp` | | | |
| Local Gradient | 87.4%
(297 / 647) | 6 334.1
(177 123) | 0.4s
(19.2s) |
| Enclosure Gradient (Ours) | **89.2**%
(311 / 652) | **1 478.1**
(138 367) | **0.3s**
(14.9s) |

## C  APPENDIX – ADDITIONAL RESULTS

In Tab. 5, we provide the results for the remaining benchmarks from Tab. 2. We briefly list our observations.

**Reduced Number of Subproblems**   Our input refinement reduces the number of subproblems across all benchmarks.

**Verification Time**   Our input refinement speeds up verification for all benchmarks except `acasxu`, `linearizenn`, and `tllverifybench`; these benchmarks have low-dimensional input spaces (Brix et al., 2024, Tab. 3), i.e., 5-dimensional for `acasxu`, 4-dimensional for `linearizenn`, and 2-dimensional for `tllverifybench`. For `acasxu` and `linearizenn`, naive input splitting already achieves fast verification times; for `tllverifybench`, the input refinement only marginally reduces the number of subproblems, and thus the computational overhead increases the overall verification time.

**Computational Overhead**   Quantifying the computational overhead of our input refinement is complicated due to batch-wise computations with varying batch sizes. Therefore, we report the maximum per-iteration time to estimate the overhead. Generally, the per-iteration time is larger with our input refinement (✓). However, for the high-dimensional benchmarks e.g., `collins-rul-cnn`, `cora`, `dist-shift`, and `metaroom`, the input refinement significantly reduces the splitting, thus, the batch sizes are significantly smaller and hence the maximum per-iteration time with input refinement is smaller; e.g., for `collins-rul-cnn`, the input refinement reduces the maximum number of subproblems from 97 (without refinement, ✗) to 2 (with refinement, ✓). Therefore, the maximum per-iteration time is reduced from 2 458.9ms (without refinement, ✗) to 86.6ms (with refinement, ✓).

## D  APPENDIX – IMPLEMENTATION TRICKS

We efficiently implement our algorithm with only matrix operations to take full advantage of GPU acceleration. Furthermore, to make our algorithm practical, we reduce the memory footprint by storing only the bounds of the constrained zonotopes in the queue. We use zonotopes for the set propagation (line 6). However, we compute the bounds of a constrained zonotope before an enqueue operation.

Computing the bounds of constrained zonotopes requires solving two linear programs for each dimension, which limits GPU acceleration. Hence, we avoid solving linear programs by efficiently approximating the bounds of a constrained zonotope by approximating the bounds of the constrained hypercube (Prop. 3). The approximation is inspired by the Fourier-Motzkin elimination (Schrijver, 1998, Sec. 12.2).

Table 5: Additional Results: without (✗) vs. with (✓) Input Refinement.

| Input Refinement | %Solved ↑ (#Verified / #Falsified) | avg. #Sub-problems ↓ (max) | avg. Time ↓ (max) | max Time per Iter. ↓ |
|---|---|---|---|---|
| acasxu | | | | |
| ✗ | **99.5%** (138 / 47) | 1611.2 (133438) | **0.7**s (32.5s) | **118.6**ms |
| ✓ (Ours) | **99.5%** (138 / 47) | **651.5** (36134) | 0.8s (15.1s) | 302.8ms |
| collins-rul-cnn | | | | |
| ✗ | **100.0%** (30 / 32) | 3.5 (97) | 8.6s (241.7s) | 2458.9ms |
| ✓ (Ours) | **100.0%** (30 / 32) | **1.5** (2) | **0.1**s (0.3s) | **86.6**ms |
| cora | | | | |
| ✗ | 40.0% (18 / 54) | 3.5 (9) | 4.9s (27.2s) | 383.3ms |
| ✓ (Ours) | **81.1%** (18 / 128) | **1** (1) | **0.1**s (0.2s) | **174.1**ms |
| dist-shift | | | | |
| ✗ | **98.6%** (63 / 8) | 64.8 (2420) | 0.6s (13.6s) | 281.4ms |
| ✓ (Ours) | **98.6%** (63 / 8) | **6.4** (229) | **0.2**s (4.9s) | **139.2**ms |
| linearizenn | | | | |
| ✗ | **100.0%** (59 / 1) | 38.0 (210) | **0.4s** (1.2s) | **50.6**ms |
| ✓ (Ours) | **100.0%** (59 / 1) | **31.5** (171) | 0.7s (1.8s) | 83.2ms |
| metaroom | | | | |
| ✗ | 93.0% (90 / 3) | 18.8 (42) | 7.6s (149.5s) | 751.2ms |
| ✓ (Ours) | **97.0%** (90 / 7) | **18.5** (26) | **1.1**s (1.9s) | **476.1**ms |
| safenlp | | | | |
| ✗ | 80.8% (298 / 575) | 4401.8 (293304) | 1.6s (19.9s) | **54.3**ms |
| ✓ (Ours) | **89.2%** (311 / 652) | **1529.2** (114065) | **0.3**s (12.4s) | 89.4ms |
| tllverifybench | | | | |
| ✗ | **100.0%** (15 / 17) | 117.6 (603) | **18.7s** (185.4s) | **2336.7**ms |
| ✓ (Ours) | **100.0%** (15 / 17) | **110.8** (595) | 36.1s (368.6s) | 5148.9ms |

To avoid clutter, we introduce further notation: For a matrix $A \in \mathbb{R}^{n \times m}$, we denote the matrix with all non-positive entries set to zero with $A^+ \in \mathbb{R}_{\geq 0}$, i.e., $A^+ = 1/2\,(A + |A|)$; the matrix $A^- \in \mathbb{R}_{\geq 0}$ is defined analogously.

**Proposition 3** (Bounds of Bounded Polytope). *A constrained hypercube $\mathcal{P} = \langle C, d \rangle_H \cap \left[\underline{\beta}_0, \overline{\beta}_0\right] \subset \mathbb{R}^q$ with $C \in \mathbb{R}^{p \times q}$ and $d \in \mathbb{R}^p$ is enclosed by $\left[\underline{\beta}, \overline{\beta}\right]$, i.e., $\mathcal{P} \subseteq \left[\underline{\beta}, \overline{\beta}\right]$. For each dimension $j \in [q]$, let $\Sigma_{\backslash\{j\}} := I_q - \mathbf{e}_j\,\mathbf{e}_j^\top$, and for each $i \in [p]$, let $\underline{C}_{(i,j)} := (d_{(i)} - C^+_{(i,\cdot)}\,\Sigma_{\backslash\{j\}}\,\underline{\beta}_0 - C^-_{(i,\cdot)}\,\Sigma_{\backslash\{j\}}\,\overline{\beta}_0)/C_{(i,j)}$. The bounds of $\mathcal{P}$ are computed as*

$$\underline{\beta}_{(j)} = \max\{\underline{C}_{(i,j)} \mid i \in [p]\colon C_{(i,j)} < 0\} \cup \{\underline{\beta}_{0(j)}\},$$
$$\overline{\beta}_{(j)} = \min\{\underline{C}_{(i,j)} \mid i \in [p]\colon C_{(i,j)} > 0\} \cup \{\overline{\beta}_{0(j)}\}.$$

*Proof.* We fix a point $\beta \in \mathcal{P}$ and indices $i \in [p]$ and $j \in [q]$ and show that $\beta \in \left[\underline{\beta}, \overline{\beta}\right]$. From the definition of a polytope, it holds that: $C_{(i,\cdot)} \beta \leq d_{(i)}$. By rearranging the terms, we obtain

$$C_{(i,\cdot)} \beta \leq d_{(i)} \quad \iff \quad C_{(i,j)} \beta_{(j)} \leq d_{(i)} - \sum_{k \in [q] \setminus \{j\}} C_{(i,k)} \beta_{(k)} = d_{(i)} - C_{(i,\cdot)} \Sigma_{\setminus \{j\}} \beta.$$

We split cases on the sign of $C_{(i,j)}$:

*Case 1 ($C_{(i,j)} < 0$).* We obtain a lower bound for $\beta_{(j)}$:

$$\frac{d_{(i)} - C_{(i,\cdot)} \Sigma_{\setminus \{j\}} \beta}{C_{(i,j)}} \leq \beta_{(j)}.$$

Using the initial bounds, i.e., $\underline{\beta}_{0(j)} \leq \beta_{(j)} \leq \overline{\beta}_{0(j)}$, we can approximate the bounds of $\beta_{(j)}$:

$$\beta_{(j)} \overset{(C_{(i,j)}<0)}{\geq} \frac{d_{(i)} - C_{(i,\cdot)} \Sigma_{\setminus \{j\}} \beta}{C_{(i,j)}} \geq \min_{\tilde{\beta} \in \mathcal{P}} \frac{d_{(i)} - C_{(i,\cdot)} \Sigma_{\setminus \{j\}} \tilde{\beta}}{C_{(i,j)}} \geq \min_{\tilde{\beta} \in [\underline{\beta}_0, \overline{\beta}_0]} \frac{d_{(i)} - C_{(i,\cdot)} \Sigma_{\setminus \{j\}} \tilde{\beta}}{C_{(i,j)}}$$

$$= \frac{d_{(i)} - \min_{\tilde{\beta} \in [\underline{\beta}_0, \overline{\beta}_0]} C_{(i,\cdot)} \Sigma_{\setminus \{j\}} \tilde{\beta}}{C_{(i,j)}} = \frac{d_{(i)} - C_{(i,\cdot)}^+ \Sigma_{\setminus \{j\}} \underline{\beta}_0 - C_{(i,\cdot)}^- \Sigma_{\setminus \{j\}} \overline{\beta}_0}{C_{(i,j)}} = \underline{C}_{(i,j)}.$$

*Case 2 ($C_{(i,j)} > 0$).* We obtain an upper bound for $\beta_{(j)}$:

$$\beta_{(j)} \leq \frac{d_{(i)} - C_{(i,\cdot)} \Sigma_{\setminus \{j\}} \beta}{C_{(i,j)}}.$$

Using the initial bounds, i.e., $\underline{\beta}_{0(j)} \leq \beta_{(j)} \leq \overline{\beta}_{0(j)}$, we can approximate the bounds of $\beta_{(j)}$:

$$\beta_{(j)} \overset{(C_{(i,j)}>0)}{\leq} \frac{d_{(i)} - C_{(i,\cdot)} \Sigma_{\setminus \{j\}} \beta}{C_{(i,j)}} \leq \max_{\tilde{\beta} \in \mathcal{P}} \frac{d_{(i)} - C_{(i,\cdot)} \Sigma_{\setminus \{j\}} \tilde{\beta}}{C_{(i,j)}} \leq \max_{\tilde{\beta} \in [\underline{\beta}_0, \overline{\beta}_0]} \frac{d_{(i)} - C_{(i,\cdot)} \Sigma_{\setminus \{j\}} \tilde{\beta}}{C_{(i,j)}}$$

$$= \frac{d_{(i)} - \min_{\tilde{\beta} \in [\underline{\beta}_0, \overline{\beta}_0]} C_{(i,\cdot)} \Sigma_{\setminus \{j\}} \tilde{\beta}}{C_{(i,j)}} = \frac{d_{(i)} - C_{(i,\cdot)}^+ \Sigma_{\setminus \{j\}} \underline{\beta}_0 - C_{(i,\cdot)}^- \Sigma_{\setminus \{j\}} \overline{\beta}_0}{C_{(i,j)}} = \underline{C}_{(i,j)}.$$

Each row $i$ of the constraint matrix $C$ generates an inequality for each dimension $j$ of $\beta$. Thus, with $\beta \in \left[\underline{\beta}_0, \overline{\beta}_0\right]$, we set $\underline{\beta}_{(j)}$ to maximum lower bound and $\overline{\beta}_{(j)}$ to minimum lower bound. $\qquad \square$

Note that Prop. 3 can be applied iteratively: For $\mathcal{P} = \langle C, d \rangle_H \cap \left[\underline{\beta}_0, \overline{\beta}_0\right]$ with bounds $\left[\underline{\beta}, \overline{\beta}\right]$, it holds that $\mathcal{P} = \langle C, d \rangle_H \cap \left[\underline{\beta}, \overline{\beta}\right]$; thus, Prop. 3 can be applied again to obtain potentially smaller bounds. Fig. 4 illustrates the approximated bounds and exact bounds of a bounded polytope.

The bounds of a constrained zonotope $\mathcal{Z}|_{C \leq d} \subset \mathbb{R}^n$ with $\mathcal{Z} = \langle c, G \rangle_Z$ can be approximated by applying an affine map to the bounds $\left[\underline{\beta}, \overline{\beta}\right] \subseteq \overline{\mathcal{B}}_q$ of its constrained hypercube:

$$\mathcal{Z}|_{C \leq d} \subseteq c \oplus \left[G^+ \underline{\beta} - G^- \overline{\beta}, G^+ \overline{\beta} - G^- \underline{\beta}\right]. \tag{13}$$

In Tab. 6, we evaluate the effects of our fast bounding a constraint zonotope (Prop. 3) on the verification accuracy. We turned off the timeout because, with the exact bounds, we could not verify any instance within the allotted time. Both approaches can verify all instances, and as expected, computing the exact bounds is significantly slower; i.e., the average verification time is 823.3s vs. 1.0s for our fast bound approximation. Furthermore, the exact bounds only marginally reduce the number of verified subproblems, i.e., on average 276.98 vs. 277.58 for the fast bound approximation, indicating that our approximated bounds are sufficiently tight for verifying neural networks.

---

[2]Computing the exact bounds of a high-dimensional constrained zonotope is computationally expensive; thus, we can only compare our approximation on the first 45 instances of the `acasxu` benchmark (`prop1`).

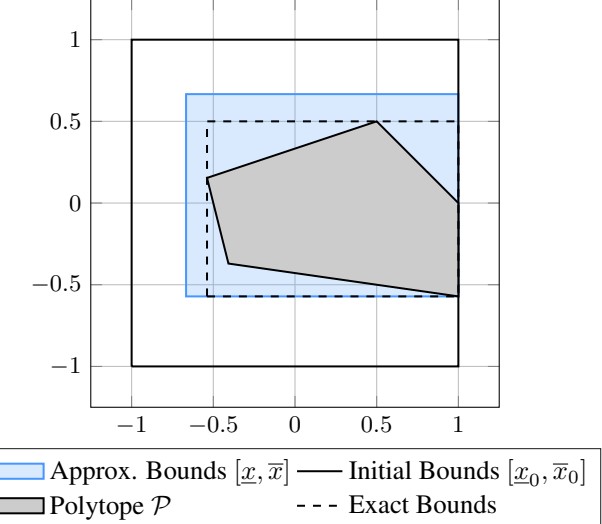

Figure 4: Illustrating the approximated bounds (Prop. 3) and the exact bounds of a bounded polytope.

Table 6: Approximation of Constraint Zonotope Bounds.

| Exact Bounds | %Solved ↑ (#Verified / #Falsified) | avg. #Sub- problems ↓ (max) | avg. Time ↓ (max) | max. Time per Iter. ↓ |
|---|---|---|---|---|
| acasxu² | | | | |
| ✓ | 100.0% (45 / 0) | 277.0 (1 045) | 823.3s (3 088.1s) | 171 559.7ms |
| ✗ (Prop. 3) | 100.0% (45 / 0) | 277.6 (1 051) | **1.0s** (1.9s) | **110.2ms** |

# E    APPENDIX – COMPUTATIONAL COMPLEXITY

The following proposition provides the time complexity of an iteration of Alg. 1.

**Proposition 4.** *Let $\kappa \in \mathbb{N}$ be the number of layers, $p \in \mathbb{N}$ be the number of output constraints, $q \in \mathbb{N}$ be the number of generators of the output enclosure, and $\xi \in \mathbb{N}$ be the number of splits in each iteration. An iteration of Alg. 1 (lines 4–21) takes $\mathcal{O}(\kappa q^3 + \xi p q^2)$ computation time with respect to $\kappa$, $p$, $q$, and $\xi$.*

*Proof.* Computing the enclosure of the output set (line 6) takes time $\mathcal{O}(\kappa q^3)$ (Koller et al., 2025, Prop. 16). Checking if the intersection of the enclosure with the unsafe set is empty (line 7) takes time $\mathcal{O}(p n_\kappa q)$. Generating a candidate counterexample (line 10) takes time $\mathcal{O}((n_0 + n_\kappa) q)$; checking the counterexample (line 11) takes time $\mathcal{O}(\kappa q^2 + p n_\kappa)$. Splitting the input set into $\xi$ new sets and refining the each new input set (lines 15–16) takes time $\mathcal{O}(\xi (p n_\kappa q + p q^2)) = \mathcal{O}(\xi p q^2)$.

The dequeue and enqueue operations take constant time (Knuth, 1997, Sec. 2.2.1).

In total, an iteration of Alg. 1 takes time $\mathcal{O}(\kappa q^3 + \xi p q^2)$. $\qquad\qquad\qquad\qquad\qquad\qquad\square$

# F    APPENDIX – PROOFS

**Proposition 2.** *Given are a neural network $\Phi \colon \mathbb{R}^{n_0} \to \mathbb{R}^{n_\kappa}$, an input set $\mathcal{X} = \langle c_x, G_x \rangle_Z \subset \mathbb{R}^{n_0}$ with $G_x \in \mathbb{R}^{n_0 \times q_0}$, and an unsafe set $\mathcal{U} = \langle A, b \rangle_H \subset \mathbb{R}^{n_\kappa}$. Let $\mathcal{Y} = \langle c_y, G_y \rangle_Z = \mathtt{enclose}(\Phi, \mathcal{X})$ be an enclosure of the output set with $G_y \in \mathbb{R}^{n_\kappa \times q_\kappa}$. We enclose all unsafe inputs by constraining $\mathcal{X}$ to $\mathcal{X}|_{C \leq d}$, i.e.,*

$$\{x \in \mathcal{X} \mid \Phi(x) \in \mathcal{U}\} \subseteq \mathcal{X}|_{C \leq d} \subseteq \mathcal{X},$$

*where $C := A\,G_{y(\cdot,[q_0])}$ and $d := b - A\,c_y + \big|A\,G_{y(\cdot,[q_\kappa]\setminus[q_0])}\big|\,\mathbf{1}$.*

*Proof.* We fix an unsafe input $x \in \{x \in \mathcal{X} \mid \Phi(x) \in \mathcal{U}\}$. Let $y = \Phi(x)$ be its corresponding output. We use the definition of the unsafe set $\mathcal{U}$ and the definition of a zonotope, i.e., there are factors $\beta \in \mathcal{B}_{q_\kappa}$ s.t.

$$y = c_y + G_y\,\beta. \tag{14}$$

For an unsafe output, we have the following inequality:

$$y \in \mathcal{U} \stackrel{(14)}{\iff} A\,(c_y + G_y\,\beta) \leq b \iff A\,G_y\,\beta \leq b - A\,c_y.$$

Please note that the first $q_0$ factors of the output $y$ are the factors of the input $x$, i.e., $x = c_x + G_x\,\beta_{([q_0])}$. Our goal is to constrain the factors $\beta_{([q_0])}$. Therefore, we rearrange the terms

$$
\begin{aligned}
A\,G_y\,\beta \leq b - A\,c_y \iff\ & A\,G_{y(\cdot,[q_0])}\,\beta_{([q_0])} + A\,G_{y(\cdot,[q_\kappa]\setminus[q_0])}\,\beta_{([q_\kappa]\setminus[q_0])} \leq b - A\,c_y \\
\iff\ & \underbrace{A\,G_{y(\cdot,[q_0])}}_{=:C}\,\beta_{([q_0])} \leq b - (A\,c_y + A\,G_{y(\cdot,[q_\kappa]\setminus[q_0])}\,\beta_{([q_\kappa]\setminus[q_0])}) \\
\iff\ & C\,\beta_{([q_0])} \leq b - (A\,c_y + A\,G_{y(\cdot,[q_\kappa]\setminus[q_0])}\,\beta_{([q_\kappa]\setminus[q_0])}).
\end{aligned}
$$

We bound the right-hand side to get to obtain constraints $C\,\beta_{([q_0])} \leq d$ on the factors of $x$:

$$C\,\beta_{([q_0])} \leq b - (A\,c_y + A\,G_{y(\cdot,[q_\kappa]\setminus[q_0])}\,\beta_{([q_\kappa]\setminus[q_0])}) \leq b - A\,c_y + \big|A\,G_{y(\cdot,[q_\kappa]\setminus[q_0])}\big|\,\mathbf{1} = d.$$

Thus, we conclude that the input $x$ is contained in the constrained input set $\mathcal{X}$:

$$x \in \mathcal{X}|_{C \leq d}. \qquad \square$$

The following corollary proves the soundness of our refinement for neural network verification and falsification.

**Corollary 1.** *Given a neural network $\Phi\colon \mathbb{R}^{n_0} \to \mathbb{R}^{n_\kappa}$, an input set $\mathcal{X} \subset \mathbb{R}^{n_0}$, and unsafe set $\mathcal{U} = \langle A, b\rangle_H \subset \mathbb{R}^{n_\kappa}$, the refinement of the input set (Prop. 2) can be used for verification and falsification*

$$
\begin{aligned}
\Phi(\mathcal{X}|_{C \leq d}) \cap \mathcal{U} = \emptyset &\iff \Phi(\mathcal{X}) \cap \mathcal{U} = \emptyset, \\
\exists x \in \mathcal{X}|_{C \leq d}\colon \Phi(x) \in \mathcal{U} &\iff \exists x \in \mathcal{X}\colon \Phi(x) \in \mathcal{U}.
\end{aligned}
$$

*Proof.* With Prop. 2 we obtain

$$\mathcal{X} \setminus \mathcal{X}|_{C \leq d} \subseteq \{x \in \mathcal{X} \mid \Phi(x) \notin \mathcal{U}\},$$

and therefore, we have

$$\Phi(\mathcal{X} \setminus \mathcal{X}|_{C \leq d}) \cap \mathcal{U} = \emptyset, \qquad \neg(\exists x \in \mathcal{X} \setminus \mathcal{X}|_{C \leq d}\colon \Phi(x) \in \mathcal{U}). \tag{15}$$

Thus,

$$
\begin{aligned}
\Phi(\mathcal{X}|_{C \leq d}) \cap \mathcal{U} = \emptyset \stackrel{(15)}{\iff}\ & \Phi(\mathcal{X}|_{C \leq d}) \cap \mathcal{U} = \emptyset \wedge \Phi(\mathcal{X} \setminus \mathcal{X}|_{C \leq d}) \cap \mathcal{U} = \emptyset \\
\iff\ & (\Phi(\mathcal{X}|_{C \leq d}) \cup \Phi(\mathcal{X} \setminus \mathcal{X}|_{C \leq d})) \cap \mathcal{U} = \emptyset \\
\iff\ & \Phi(\mathcal{X}) \cap \mathcal{U} = \emptyset,
\end{aligned}
$$

$$
\begin{aligned}
\exists x \in \mathcal{X}|_{C \leq d}\colon \Phi(x) \in \mathcal{U} \stackrel{(15)}{\iff}\ & \exists x \in \mathcal{X}|_{C \leq d}\colon \Phi(x) \in \mathcal{U} \vee \exists x \in \mathcal{X} \setminus \mathcal{X}|_{C \leq d}\colon \Phi(x) \in \mathcal{U} \\
\iff\ & \exists x \in (\mathcal{X}|_{C \leq d} \cup \mathcal{X} \setminus \mathcal{X}|_{C \leq d})\colon \Phi(x) \in \mathcal{U} \\
\iff\ & \exists x \in \mathcal{X}\colon \Phi(x) \in \mathcal{U}. \qquad \square
\end{aligned}
$$

## G  DISCLOSURE – USAGE OF LARGE LANGUAGE MODELS (LLMs)

We used a large language model (LLM) as a general-purpose tool to refine the writing and enhance clarity of expression. All research ideas, methodology, analysis, and conclusions are solely the work of the authors.

