# OpenReview forum: "Out of the Shadows: Exploring a Latent Space for Neural Network Verification"
_ICLR.cc/2026/Conference — ICLR 2026 Poster_

### Official Review · Reviewer_JVYp · 2025-10-28

**Soundness:** 4
**Presentation:** 3
**Contribution:** 4
**Rating:** 8
**Confidence:** 2

**Summary:**

This paper introduces a novel latent-space formulation for neural network verification based on projection-based set representations, such as zonotopes. The key idea is to interpret the sets propagated through a network as “shadows” of a shared higher-dimensional latent set, which enables transferring output constraints back to the input domain for iterative specification-driven input refinement. The proposed method integrates this refinement into a branch-and-bound verification framework, implemented purely with matrix operations for efficient GPU acceleration. Experimental results on VNN-COMP’24 benchmarks show competitive performance and substantial reductions in subproblems compared with state-of-the-art tools.

**Strengths:**

1.Conceptual originality -The latent-space interpretation of set propagation provides a fresh geometric perspective on verification problems.
2.Technical soundness-The integration of constrained zonotopes and iterative refinement is mathematically consistent and grounded in established reachability theory.
3.Practical scalability-The matrix-only implementation with GPU acceleration is elegant and demonstrates a real performance gain.
4.Comprehensive evaluation -The comparison with multiple VNN-COMP tools and detailed ablations add credibility to the empirical claims.
5.Potential for future extensions-The framework could generalize to other activation functions and verification domains (e.g., control systems).

**Weaknesses:**

Clarification of Competition Reference： The paper states that the proposed tool would rank among the top of VNN-COMP’24, which, while illustrative, might risk breaking double-blind policy if the authors were participants or submitted related work. A clearer phrasing could avoid potential self-identification.

Limited Discussion on Generalization： While the latent-space formulation elegantly improves efficiency, its scalability to larger architectures beyond benchmarks (e.g., transformers or diffusion models) is not discussed, leaving some uncertainty about general applicability.

Evaluation Scope： Experiments are strong on VNN-COMP tasks, yet cross-domain or real-world safety-critical benchmarks (e.g., perception-driven models) could further demonstrate practical robustness. Overall, these are minor concerns rather than major flaws.

**Questions:**

1.How does the proposed latent-space refinement behave when using non-piecewise-linear activations (e.g., tanh, sigmoid)? Can the method generalize without losing soundness guarantees?
2.The refinement procedure iteratively constrains inputs based on unsafe output sets. How is termination ensured in practice—by a fixed iteration limit or a convergence threshold?
3.Since the algorithm uses only matrix operations, could the authors discuss numerical stability issues (e.g., accumulation of floating-point errors) in deep networks?
4.Double-Blind Compliance: The manuscript notes that the proposed verifier “would place it among the top-ranking tools of the last VNN-COMP’24.” Could the authors clarify whether they personally participated in VNN-COMP’24 or reused their own competition submissions? If so, this phrasing might unintentionally compromise anonymity. A general comparative statement without potential self-identification could be safer under ICLR’s double-blind policy.
5.It would be valuable to understand the trade-off between refinement accuracy and computational overhead—does aggressive refinement ever hurt verification completeness?

---

> ### Author Response · Authors · 2025-11-18
>
> Dear reviewer JVYp,
>
> Thank you for your thoughtful comments and helpful suggestions. For your convenience, we mark the changes in our revised paper in blue.
>
> **Generalization and Scalability:**
> While our experiments focus on VNN-COMP benchmarks, these include several high-dimensional convolutional architectures (e.g., $\texttt{collins-rul-cnn}$, $\texttt{metaroom}$) that demonstrate our input refinement improves the verification of large neural networks.
> Furthermore, the "shadow" view is not limited to specific architectures and could theoretically be extended to more complex architectures, e.g., transformers.
>
> **Evaluation on Safety-Critical Tasks:**
> We agree that evaluating the approach on a broader range of real-world safety-critical settings would be valuable for future work. We would like to highlight that the already included $\texttt{acasxu}$ benchmark directly addresses the safety verification of neural networks used in the Airborne Collision Avoidance System (ACAS).
>
> **1. Generalization to non-ReLU networks:**
> Yes, our verification algorithm can already handle non-ReLU activations, i.e., using a linear approximation and corresponding error terms. Thus, the "shadow" view works for non-ReLU activations, because we still enclose each layer using a projection of the same latent space, and the input refinement remains sound. Please note that the $\texttt{dist-shift}$ benchmark already uses neural networks with sigmoid activations. We clarified this generality in the revised paper.
>
> **2. Input Refinement -- Termination:** In all experiments, we apply a fixed number of refinement iterations: In each iteration of Alg. 1, we apply 8 refinement iterations to the split input sets. In practice, this provides a good balance between accuracy and overhead. We clarify this in Appendix A.
>
> **3. Numerical Stability:** In our experiments, we did not observe any additional numerical instability caused by our input refinement. To clarify this, we added a short discussion about numerical stability to Appendix A.
>
> **4. Double-Blind Compliance:** Thank you for raising this important point regarding double-blind compliance; we agree with the sensitivity of the issue. We can confirm that the proposed approach is novel and did not participate in VNN-COMP'24.
>
> **5. Input Refinement -- Overhead:** Thank you for raising this. We expanded the evaluation in Sec. 5.1 to quantify the computational trade-offs, with timing data for all remaining benchmarks in Appendix C.
> Across benchmarks, we observe that input refinement typically increases per-iteration time, as expected, but significantly reduces the number of subproblems, which results in lower overall verification time.
>
> Moreover, our input refinement does not affect completeness, because only safe input regions are discarded; thus, our input refinement improves overall performance without harming soundness or completeness.

---

### Official Review · Reviewer_hneQ · 2025-10-31

**Soundness:** 3
**Presentation:** 4
**Contribution:** 3
**Rating:** 6
**Confidence:** 2

**Summary:**

This paper proposes a novel refinement method for zonotope-based verification of the robustness properties of neural networks.

Specifically, it describes a latent space, meaning a hypercube, with the property that the input zonotope, as well as the intermediate and output zonotopes while and after propagating through the network, are all "shadows" of this hypercube, that is, they are projections. This method enables the creation of a relatively tight enclosure of the set of inputs.

The performance of the proposed procedure is demonstrated experimentally on SOTA benchmarks by incorporating the proposed refinement into a branch-and-bound algorithm. The experiments show that it is often competitive
with existing high-performing ones verifiers. Additionally, the number of subproblems that
must be solved is, in some cases, significantly fewer than without the proposed refinement.

**Strengths:**

Improving the performance of FNN verifiers, particularly those based on zonotopes, is a topic of significant relevance and a suitable contribution for ICLR.

Furthermore, the paper is excellently written and thoroughly polished,
making it a pleasure to read. This is particularly evident in the balance
between formal rigour and the intuitive explanations provided by the authors.

**Weaknesses:**

I identify one notable weakness:
- The experimentally demonstrated improvements, while present, are not dramatic. I perceive the incremental enhancement offered by this refinement as a valuable contribution. Nonetheless, as I am not a specialist in this precise domain, I wonder if the enhancement might be too modest or if further extensive experiments are required to substantiate the minor, yet existent, improvements due to this refinement approach.

Minor issues include:
- The explanation of "shadows" as a concept of refinement is appreciated. However, while it is clear that the latent hypercube remains the same across steps but is rotated (see, for instance, Figure 2), the connection between this idea and Proposition 2 is not entirely clear. A slightly clearer exposition could be beneficial.
- Although it is not central to the paper, the introduction's claim that "verification of FNN is in general undecidable" and "NP-complete for ReLU FNN" feels superficial. There is not a single "verification problem" for FNNs. Various notions exist, such as robustness and reachability, each contingent on parameters like input and output set specifications and FNN architecture types. Consequently, this has spawned extensive literature analysing the complexity of these problems in detail (see, e.g., [1],[2],[3] and others). Furthermore, the undecidability assertion appears misplaced. While referenced in another paper, so not a statement of the authors, it arises simply from the existence of undecidable sets of reals and is not an inherent property of "FNN verification" problems.
- Proposition 2: The definition of $d:=...$ is unclear to me. This may be due to a typo.

---
[1] Marco Sälzer, Martin Lange:
Reachability in Simple Neural Networks. Fundam. Informaticae 189(3-4): 241-259 (2022)

[2] Adrian Wurm:
Robustness Verification in Neural Networks. CPAIOR (2) 2024: 263-278

[3] Moritz Stargalla, Christoph Hertrich, Daniel Reichman:
The Computational Complexity of Counting Linear Regions in ReLU Neural Networks. CoRR abs/2505.16716 (2025)

**Questions:**

- I am curious about the reasonable next steps for the approach to refinement you have proposed here. More specifically, I am attempting to understand whether this "novel view" you propose paves the way for follow-up works and, if so, in what manner?

---

> ### Author Response · Authors · 2025-11-18
>
> Dear reviewer hneQ,
>
> Thank you very much for your time and your valuable comments. For your convenience, we mark the changes in our revised paper in blue.
>
> **Experimental Improvements:** We have significantly expanded the evaluation with additional runtime data and per-iteration time (Tab. 2b and Appendix C) to more clearly show the gains from our contributions: (i) our input refinement and (ii) full GPU acceleration. (i) Across benchmarks, we observe that our input refinement typically increases per-iteration time, as expected, but significantly reduces the number of subproblems, which results in lower overall verification time. (ii) The GPU acceleration speeds up the verification by 89.8% on safenlp compared to CPU-based verification.
>
> **Connection between Ex. 1 and Prop. 2:** The latent space (hypercube) is visualized in Ex. 1 and enables transfer of output specifications to the input space, which motivates our input refinement. We revised Ex. 1 and Prop. 2 to better illustrate their connection.
>
> **Undecidability of Neural Network Verification:** We appreciate this observation. The focus of our paper is the standard safety verification of neural networks (Sec. 2.5): determine whether a neural network maps any input in a given set into an unsafe output set. This problem is known to be undecidable for general networks, and NP-complete for ReLU feedforward networks.
>
> **Follow-Up Works:** Thank you for asking about follow-up work. Our current method refines only the input set with respect to an output specification. A natural next step is to extend the refinement mechanism to intermediate layers, e.g., leveraging constraints to refine the enclosure of activation functions.

---

> > ### Comment · Reviewer_hneQ · 2025-11-25
> >
> > Thanks for adressing my concerns and question. I have no follow-up questions.

---

### Official Review · Reviewer_jJ3V · 2025-10-31

**Soundness:** 4
**Presentation:** 2
**Contribution:** 3
**Rating:** 6
**Confidence:** 4

**Summary:**

This paper presents a novel verification algorithm for neural networks that integrates an iterative, specification-driven input refinement into a branch-and-bound procedure. The core idea is to use the dependency information preserved by zonotope propagation to "transfer" output constraints back to the input space, thereby refining the input set to focus only on potentially unsafe inputs. The implementation is designed for full GPU acceleration via matrix operations. The evaluation on VNN-COMP'24 benchmarks demonstrates competitive, state-of-the-art performance, with extensive ablation studies validating the design choices.
However, the paper's central conceptual framing of a "novel latent space" is overstated, as it repackages the standard mathematical interpretation of zonotopes without providing a meaningful new insight for the verification task. Furthermore, the analysis lacks a thorough discussion of the computational overhead introduced by the iterative refinement procedure itself.

**Recommendation:**
Despite the conceptual overreach regarding the "latent space," the paper presents a solid, practical, and effective verification algorithm with strong empirical results. The core technical contribution of the iterative input refinement is novel and valuable. Given the competitive performance and efficient implementation, I am inclined towards **acceptance**, provided the authors significantly revise the framing to more accurately reflect the nature of their contribution and address the points above.

**Strengths:**

1.  **Novel and Sound Method:** The iterative input refinement procedure (Proposition 2) is a novel and technically sound contribution. By iteratively constraining the input set based on the unsafe output specification, the method effectively reduces the number of subproblems in the branch-and-bound tree, leading to faster verification.
2.  **Efficient Implementation:** The decision to implement the algorithm using only matrix operations to leverage full GPU acceleration is a significant practical strength. The results show a substantial speed-up compared to CPU execution, making the approach scalable.
3.  **Convincing Evaluation:** The experimental evaluation is comprehensive and transparent. The tool achieves competitive performance against the top-5 tools from VNN-COMP'24. The ablation studies effectively demonstrate the individual contributions of the input refinement, GPU acceleration, and the proposed enclosure-gradient splitting heuristic.
4.  **Practical Impact:** The method leads to tangible improvements in verification performance, including a reduced number of subproblems, faster solve times, and a high success rate across diverse benchmarks.

**Weaknesses:**

1.  **Overstated Framing and Novelty of "Latent Space":** The paper's central conceptual framing—a "novel latent space" for verification—is overstated. The observation that a zonotope is a projection of a hypercube is a standard, well-known interpretation of the zonotope definition. Maintaining a single, constant hypercube by padding generator matrices with zeros is an algebraic convenience rather than a profound theoretical insight. The paper fails to provide a meaningful interpretation of this "latent space" for the verification task itself. Consequently, the "latent space" narrative feels more like a post-hoc metaphor than a foundational concept that the algorithm genuinely depends upon in a deep way. The core algorithmic contribution (the refinement procedure) could be clearly explained without this conceptual overhead.
2.  **Incomplete Algorithm Description:** The presentation of the core algorithm (Algorithm 1) relies heavily on appendices for critical components, including the adversarial input computation (Section 4.2) and constraint computation (Proposition 2 and Appendix B). This makes it difficult to understand the complete method from the main text alone.
3.  **Limited Analysis of Refinement Overhead:** While the paper demonstrates that refinement reduces the number of subproblems, it doesn't adequately analyze the computational cost of the refinement procedure itself. The trade-off between reduced branching and increased per-iteration complexity deserves more discussion.

### Minor Issues
- Typo: P3 L149: $\beta\in [-1,1]^{q}$
- Typo: P5 L262: constrain
- I had problems understanding the statement of Proposition 2 because of the way it is formulated. Clarify that "$\mathcal{X}|_{C\leq d}$ encloses the intersection of the input set with the preimage of the unsafe set" is the main statement.

**Questions:**

*   Could the authors clarify what is gained by the "latent space" interpretation beyond providing an intuitive name for the shared factor space? Does it lead to algorithmic insights that are not apparent from a standard zonotope dependency analysis?
*   The paper should temper its claims regarding the novelty of the latent space view and reframe the contribution to more accurately highlight the effective iterative refinement procedure and its efficient GPU implementation as the primary advances.
*   Could the authors provide more analysis of the computational overhead of the refinement procedure and its impact on overall verification performance?
*   The algorithm description would benefit from including more critical details in the main text, particularly for the adversarial input computation and constraint generation steps.

---

> ### Author Response · Authors · 2025-11-18
>
> Dear reviewer jJ3V,
>
> Thank you for your thoughtful and detailed feedback. We appreciate the constructive suggestions and have addressed them as follows. Additionally, we have rephrased Prop. 2 to improve clarity. For your convenience, we mark the changes in our revised paper in blue.
>
> **Framing and Novelty:** We agree that the fact that zonotopes are projections of hypercubes is a standard and immediate result from the definition of zonotopes. Our intention was not to claim novelty for this part, but to provide a visual view of the single shared factor space of the entire propagation chain that serves as an intuitive motivation for our novel specification-driven input refinement.
>
> We have significantly revised the abstract, introduction, and conclusion of the paper to temper our claims of conceptual novelty and to emphasize that our primary contributions are (i) the input refinement procedure and (ii) its efficient GPU implementation.
>
> **Description of Alg. 1:** We expanded the description of Alg. 1 with details on the refinement and constraint generation. We intentionally keep the detailed bounding procedure in Appendix D to avoid burdening Alg. 1 with domain-specific implementation details.
>
> **Input Refinement Overhead:** We added a per-iteration complexity analysis of our main algorithm (Alg. 1), including the input refinement step (Appendix D), and expanded the evaluation in Sec. 5.1 to quantify the computational trade-offs, with additional timing results in Appendix C.
>
> **"Latent Space" Interpretation:** The latent space provides a unified geometric view of all propagated sets as projections of a single object, making inter-layer dependencies visually and algebraically explicit. This perspective motivates and simplifies our input refinement; however, we now present it as a visual motivation, not as the central contribution.

---

> > ### Comment · Reviewer_jJ3V · 2025-11-26
> >
> > Thank you for your thorough response and the revisions made to the manuscript, especially reframing the contributions and adding the requested analysis on computational overhead.
> >
> > I have no further questions and I will consider the changes in my final rating.

---

### Official Review · Reviewer_Lqdf · 2025-11-03

**Soundness:** 3
**Presentation:** 3
**Contribution:** 3
**Rating:** 6
**Confidence:** 5

**Summary:**

The paper proposes a latent-space framework for abstract-domain propagation in neural network verification. The main idea is that instead of treating each layer's abstract element separately, the authors view them as projections of a higher-dimensional hypercube. This allows the method to "pull back" unsafe output constraints into the latent space and iteratively refine the input set based on those constraints. The approach integrates with a branch-and-bound (BnB) verifier and is designed to run efficiently on GPUs. Experimental results on VNN-COMP'24 benchmarks show competitive performance with fewer subproblems and good results.

The paper proposes a creative latent-space formulation for zonotope-based verification that conceptually unifies input and output abstractions and enables input refinement. While the idea is novel and shows promise for improving scalability, clarification of the soundness of the refinement and further exploration of the empirical impact is needed. Overall, the contribution is original and technically interesting.

**Strengths:**

The paper introduces a clear and creative latent-space interpretation of zonotope propagation. Viewing all layer abstractions as projections of a shared higher-dimensional B-space provides an interesting way to reason about dependencies between input, hidden, and output abstractions.

Proposition 2 provides a sound theoretical basis for transferring output-space constraints back to input space, and the overall refinement process integrates cleanly within a branch-and-bound verification loop. Well written pseudocode.

The experimental evaluation covers multiple VNN-COMP'24 benchmarks and includes evaluation that isolate the effects of the proposed refinements, GPU batching, and splitting strategies. The results show tangible improvements in efficiency (fewer subproblems, faster verification) while maintaining comparable verification coverage.

Figures such as Figure 3 and the toy example make the geometric intuition accessible and help connect the abstract mathematics to practical verification behavior. Great figures overall.

**Weaknesses:**

While Proposition 2 and Corollary 1 justify the logical use of the refinement in verification, they don't prove that the latent-space refinement preserves enclosure of the reachable set. In particular, the paper does not formally establish that the refined zonotope Y_refined still over-approximates all reachable outputs of the corresponding refined input region X_refined.

Although the method is described as applicable to "projection-based" abstractions, the implementation and evaluation are restricted to zonotopes. Consider either extending the analysis to another domain (e.g., star sets) or narrowing the claim.

The paper's main conceptual innovation, the latent-space input refinement, is only briefly evaluated. Table 2(a) reports only a few benchmarks, where acasxu shows reduced subproblems but no timing improvement, and timing metrics are absent from Table 1. Expanding the evaluation to include additional benchmarks in Table 2(a) and timing data in Table 1 (even in the appendix) would clarify whether the reported efficiency gains primarily stem from the refinement itself or from implementation factors, and would enable readers to better judge the consistency and generality of the refinement's impact.

Minor Comments/Typos Found

"contrain" - > "constrain"
Missing % for some of the results of the tables.
"activiation" - > "activation"
Consistency in notation for vectors (bold vs. plain).
Table 1: define "Solved" in the caption and clarify what counts as verified/falsified.

**Questions:**

Discuss and clarify weaknesses above.

Additionally, the empirical results presented are mostly on smaller neural networks from VNN-COMP. Please clarify and discuss the capability to handle larger networks and other tasks, such as image classification. Can the approach work on adversarial robustness of MNIST, CIFAR, etc.? This would perhaps better highlight the capabilities of the approach to scale, as claimed in the contributions (and that the reviewer agrees is an interesting and potentially effective idea to exploit the latent space as an abstraction).

---

> ### Author Response · Authors · 2025-11-18
>
> Dear reviewer Lqdf,
>
> Thank you very much for your time and your valuable comments. We have corrected the typos and clarified the metrics in Tab. 1, as you pointed out. For your convenience, we mark the changes in our revised paper in blue.
>
> **Soundness of the Input Refinement:** You are correct that the refined input set does not, in general, preserve an enclosure of the full output set. We want to stress that the goal of Prop. 2 is to only enclose unsafe outputs. As proven in Cor. 1, the input refinement is sound because we only discard parts of the input that are proven to be safe, and we continue verification only of the unknown input regions. We have clarified this point at the end of Sec. 3.
>
> **Non-Zonotope Experiments:** For our implementation and evaluation we focus on zonotopes due to their favorable computational complexity and GPU-friendliness. Extending the approach to other projection-based representations, such as star sets, is a promising direction for future work.
>
> **Additional Timing Data and Computational Trade-Off:** Thank you for highlighting this. We have expanded the empirical evaluation:
>
> 1. We added timing data for the remaining benchmarks in Tab. 2a to Appendix C.
> 2. We revised Tab. 2a itself to explicitly report the maximum per-iteration time. This directly quantifies the computational overhead introduced by our input refinement.
>
> Across benchmarks, we observe that input refinement typically increases per-iteration time, as expected, but significantly reduces the number of subproblems, which results in lower overall verification time.
>
> **Scalability to larger Neural Networks (MNIST, CIFAR10):**
> In our evaluation, several benchmarks already use high-dimensional and/or convolutional architectures, e.g., $\texttt{collins-rul-cnn}$ (up to 800 input dimensions), $\texttt{cora}$ (up to 3072 input dimensions), and $\texttt{metaroom}$ (5376 input dimensions). In particular, $\texttt{cora}$ contains instances for the robustness verification of neural networks on MNIST and CIFAR10.
> Our method provides consistent improvements on these instances, supporting the claim that the latent-space refinement scales beyond small fully connected networks.

---

### Author Response · Authors · 2025-11-25

Dear reviewers,

Thank you again for your thoughtful and constructive feedback on our submission. We hope our rebuttal has addressed your questions and clarified the points you raised. If any remaining concerns are preventing an update to the score, we would be glad to provide further clarification.

Best regards,
The authors

---

### Meta-Review · Area_Chair_HDHb · 2026-01-06

**Summary:**

This paper proposes an input refinement procedure for neural network verification that leverages dependency information preserved by projection-based abstract domains, and integrates the refinement into a GPU-accelerated branch-and-bound verifier. Reviewers generally agree that the core idea is techncally sound, and that the implementation is efficient and competitive on VNN-COMP’24 benchmarks.

The main issues raised during review concerned the framing of the contribution, particularly the emphasis on a “latent space” interpretation, as well as the need for clearer justification of soundness, better analysis of computational trade-offs, and more detailed empirical reporting. While some reviewers initially viewed the gains as incremental or the conceptual framing as overstated, the technical contribution itself was regarded as solid.

**Reviewer Concerns:**

### Concerns largely addressed by the rebuttal:

Several reviewers questioned whether the “latent space” perspective constituted a novel theoretical contribution. In response, the authors appropriately revised the framing to present it as an interpretive and motivational view, rather than the core novelty, and refocused the paper on the input refinement algorithm and its efficient GPU implementation.

Questions about the soundness of the refinement procedure were clarified. The authors explained that the refinement is designed to discard only input regions that are provably safe, and therefore does not compromise soundness or completeness of verification.

Requests for clearer analysis of computational overhead were addressed through added timing data, per-iteration complexity discussion, and expanded empirical results, which helped clarify the trade-off between refinement cost and reduced branching.

Presentation issues, including metric definitions, algorithm clarity, and minor errors, were corrected in the revised manuscript.


### Concerns that remain or are only partially addressed:

The experimental evaluation is still focused on zonotopes, despite claims of applicability to broader projection-based abstractions. This limitation is acknowledged and deferred to future work.

While scalability is discussed and supported by higher-dimensional benchmarks, extension to substantially larger or different model classes remains speculative.

One reviewer noted that the empirical improvements are marginal; this is more a matter of scope and emphasis than a technical flaw.

**Reviewer Scores:**

Reviewer Lqdf: Initially at the acceptance threshold. After the clarifications on soundness and the added empirical analysis, the reviewer would likely maintain the score.

Reviewer jJ3V: Expressed satisfaction with the revisions and indicated willingness to increse the score.

Reviewer hneQ: Confirmed that concerns were addressed and had no follow-up questions.

Reviewer JVYp: 8 before the rebuttal, so no change expected.

---

### Decision · Program_Chairs · 2026-01-26

Accept (Poster)